# Acoustofluidic-based therapeutic apheresis system

Mengxi Wu [1,2,10], Zhiteng Ma [2,10], Xianchen Xu [2], Brandon Lu[3], Yuyang Gu [2], Janghoon Yoon [4], Jianping Xia [2], Zhehan Ma[3], Neil Upreti [3], Imran J. Anwar [4], Stuart J. Knechtle [4], Eileen T. Chambers[4], Jean Kwun [4] ✉, Luke P. Lee [5,6,7,8,9] ✉ & Tony Jun Huang [2] ✉

Therapeutic apheresis aims to selectively remove pathogenic substances, such as antibodies that trigger various symptoms and diseases. Unfortunately, current apheresis devices cannot handle small blood volumes in infants or small animals, hindering the testing of animal model advancements. This limitation restricts our ability to provide treatment options for particularly susceptible infants and children with limited therapeutic alternatives. Here, we report our solution to these challenges through an acoustofluidic-based therapeutic apheresis system designed for processing small blood volumes. Our design integrates an acoustofluidic device with a fluidic stabilizer array on a chip, separating blood components from minimal extracorporeal volumes. We carried out plasma apheresis in mouse models, each with a blood volume of just 280 µL. Additionally, we achieved successful plasmapheresis in a sensitized mouse, significantly lowering preformed donor-specific antibodies and enabling desensitization in a transplantation model. Our system offers a new solution for small-sized subjects, filling a critical gap in existing technologies and providing potential benefits for a wide range of patients.

Therapeutic apheresis is a medical procedure that selectively removes disease-causing agents from the blood, primarily antibodies, thus alleviating various symptoms and diseases[1]. This procedure encompasses two distinct categories: plasmapheresis, in which plasma and other non-cellular components are selectively separated, and cytapheresis, in which cellular components are selectively separated. The critical nature of therapeutic apheresis is underscored by its role as a first-line treatment for a diverse array of medical conditions, including antibody-mediated organ transplant rejection, desensitization for solid organ transplantation, immune-mediated human disorders, poisonous or drug intoxication, thrombotic thrombocytopenic purpura, sickle cell disease, chronic lymphocytic leukemia, and so on[2,3].

Effective clinical therapeutic apheresis necessitates systems characterized by high efficiency, biocompatibility, robustness, and ease of operation. Since the development of the first apheresis prototype in the 1960s, substantial efforts have been invested in the

[1]School of Mechanical Engineering, Dalian University of Technology, Dalian, Liaoning, P.R. China. [2]Thomas Lord Department of Mechanical Engineering and Materials Science, Duke University, Durham, NC 27708, USA. [3]Department of Biomedical Engineering, Duke University, Durham, NC 27708, USA. [4]Department of Surgery, Duke Transplant Center, Duke University Medical Center, Durham, NC 27708, USA. [5]Renal Division and Division of Engineering in Medicine, Department of Medicine, Harvard Medical School, Harvard University, Brigham and Women's Hospital, Boston, MA 02115, USA. [6]Department of Bioengineering, University of California, Berkeley, Berkeley, CA 94720, USA. [7]Department of Electrical Engineering and Computer Science, University of California, Berkeley, Berkeley, CA 94720, USA. [8]Department of Biophysics, Institute of Quantum Biophysics, Sungkyunkwan University, Suwon, Korea. [9]Department of Chemistry and Nanoscience, Ewha Womans University, Seoul, Korea. [10]These authors contributed equally: Mengxi Wu, Zhiteng Ma. ✉e-mail: jean.kwun@duke.edu; lplee@bwh.harvard.edu; tony.huang@duke.edu

refinement of apheresis devices[4,5]. The process is predominantly performed using automated instruments that rely on either centrifugation or membrane filtration[5,6]. These instruments include tubing sets, valves, centrifuge bowls, and filtration chambers, which demand substantial blood volumes. As a result, the standard apheresis system extracorporeal volume typically falls within the range of 165–280 mL[7].

Though this blood loss is negligible for adults with approximately 5 L of circulating blood volume, losing this quantity of blood can be life-threatening for individuals with smaller blood volumes, such as newborns. Specifically, 165 mL represents more than 50% of the total blood volume of a full-term neonate and nearly 30% of the total blood volume of a 6-month-old neonate, as per calculations following World Health Organization (WHO) guidelines[8]. This loss of blood volume exceeds the safe limits for neonatal care, making standard apheresis procedures unsuitable treatment options[9]. As a result, therapeutic apheresis in neonates is usually performed manually[10,11]. The same challenge extends to small animals as their total blood volume often closely matches (such as cats, dogs, or monkeys) or falls significantly below (such as mice, rabbits, or piglets) the 165 mL threshold. For example, the typical blood volume of a mouse ranges from 1.5 to 2.5 mL, rendering many current apheresis systems incompatible with mouse models, hindering preclinical apheresis advancements that can be translated to neonates and small children. Given the existing limitations of traditional therapeutic apheresis methods, a new apheresis system with a small extracorporeal volume is urgently needed.

The utilization of animal models is a ubiquitous practice in biomedical research, serving as a vital step in the evaluation of therapeutic strategies prior to transitioning to human clinical trials[12]. However, therapeutic apheresis has followed an inverted trajectory, whereby initial development and widespread use occurs in humans before being extended to animal subjects[13]. The exploration of therapeutic apheresis in experimental animal models has been limited due to the need for more suitable equipment. Some pioneering studies have sought to adapt existing apheresis instruments for companion animals like dogs by modifying protocols or resizing the machines[14,15]. However, these modifications introduce uncertainties regarding the therapeutic effectiveness, and there is limited flexibility in reducing the extracorporeal volume. As such, the development of a micro-apheresis system is imperative for achieving greater blood conservation. Wallukat et al. developed an aptamer column-based apheresis system to clear β1-receptor autoantibodies in rats[16], and Ma et al. developed a column-based device filled with cellulose acetate beads to conduct granulocyte and monocyte adsorptive apheresis in rats[17]. Yet, these systems still feature an extracorporeal volume of 4.5–5 mL, rendering them impractical for use with mice, the most commonly employed laboratory model species. Moreover, these attempts did not prioritize the development of a robust and user-friendly therapeutic apheresis system. The absence of such systems tailored for laboratory animal species has significantly impeded advancements in enhancing the safety and efficacy of clinical practice and conducting cost-effective, low-risk investigations to expand the applications of therapeutic apheresis in pre-clinical settings.

A microapheresis machine must delicately manage small blood volumes, separating components with minimal damage and loss while maintaining high biocompatibility. Because of the following three characteristics, acoustofluidic technology[18–40] offers a promising approach for microapheresis applications. Firstly, the acoustofluidic approach is gentle and biocompatible, preserving the integrity and functionality of biological samples and blood components post-separation[41–43]. Secondly, the acoustofluidic process is label-free and eliminates the need for specific reagents or media, thereby avoiding the introduction of unnecessary or unwarranted components into the blood and providing flexibility in utilizing various fluids such as frozen fresh plasma, albumin, or other substitute solutions. Thirdly, the acoustofluidic approach employs microfluidic channels to process

liquids, allowing for a fully integrated system that demands a smaller extracorporeal volume.

Given these considerations, our system presents the first therapeutic apheresis system for mouse models and other subjects with similarly small blood volumes. Our acoustofluidic-based therapeutic apheresis system (ATAS) facilitates precise and stable blood withdrawal and the return of prescribed blood components to the subject (i.e., mouse) while maintaining an extracorporeal volume of just 280 μL. Our work represents a reduction of three orders of magnitude compared to conventional apheresis machines and more than ten times smaller than experimental systems reported in the literature[14–17]. To accomplish this, our system features an integrated acoustofluidic chip equipped with a fluidic stabilizer, allowing for acoustic separation and stable fluid flow. This acoustofluidic method efficiently divides whole blood into plasma and cellular components with high biocompatibility, efficiency, and stability. Our ATAS permits the separation of unbound and pathogenic donor-specific antibodies from recipient blood with minimal loss of blood cells and platelets and minimal harm to the recovered blood components and the recipient animal. The ATAS platform can process blood with a throughput range of 12–18 μL min$^{-1}$, allowing the system to process the entire blood volume contained by an experimental mouse in ~2 h. Utilizing the ATAS, we experimentally validated this claim and conducted the first therapeutic apheresis procedure on a sensitized mouse model, reducing preformed donor-specific antibodies and minimizing the risk of antibody-mediated rejection following organ transplantation. Our work heralds a promising solution for conducting therapeutic apheresis on small animal models, a feat unattainable by existing technologies. This innovation unlocks new research opportunities that were previously inaccessible in small animal models, fostering advancements in apheresis-related studies. Furthermore, it paves the way for developing pediatric-specific apheresis instruments, fulfilling an unmet therapeutic need.

## Results
### Strategy and prototype of the ATAS
The acoustofluidic-based strategy for therapeutic apheresis and the prototype of ATAS is depicted in Fig. 1. This strategy was devised to address the specific demands of therapeutic apheresis procedures in clinical practice and scientific research involving subjects with small blood volume, as depicted in Fig. 1a. For example, neonates, especially preterm neonates, possess minimal total blood volume (typically around 250 mL), presenting unique challenges in neonatal intensive care units. Companion animals, such as cats, may have a total blood volume of approximately 250 mL, depending on their weight, with a dangerous threshold for blood loss being around 50 mL. Moreover, laboratory animals often used in life science research also have relatively small total blood volumes. The absence of a "microapheresis machine" has rendered therapeutic apheresis impractical in these scenarios. The primary objective of the ATAS is to overcome these obstacles in neonatal care, small animals in veterinary clinics, and life science research.

At the core of the ATAS is the acoustofluidic chip, an integrated blood microprocessor consisting of several key functions, as demonstrated in Fig. 1b. Initially, as blood or buffer enters the acoustofluidic chip via peristaltic pumps, cavity-based fluid stabilizers convert fluctuating and oscillatory flows into a stable pattern, as shown in inset (i) of Fig. 1b. The stabilizers achieve this feature by compressing and decompressing air in the cavities, ensuring constant pressure and flow rates for blood and buffer alike. Next, laminar blood focusing is achieved through sheath flow fluids (inset (ii) in Fig. 1b). A fluid splitter separates the buffer into two microchannels, which then merge as sheath channels at the junction where buffer and blood meet the main channel. This configuration utilizes the sheath flows to direct the bloodstream into the subsequent stage. Lastly, acoustic separation of

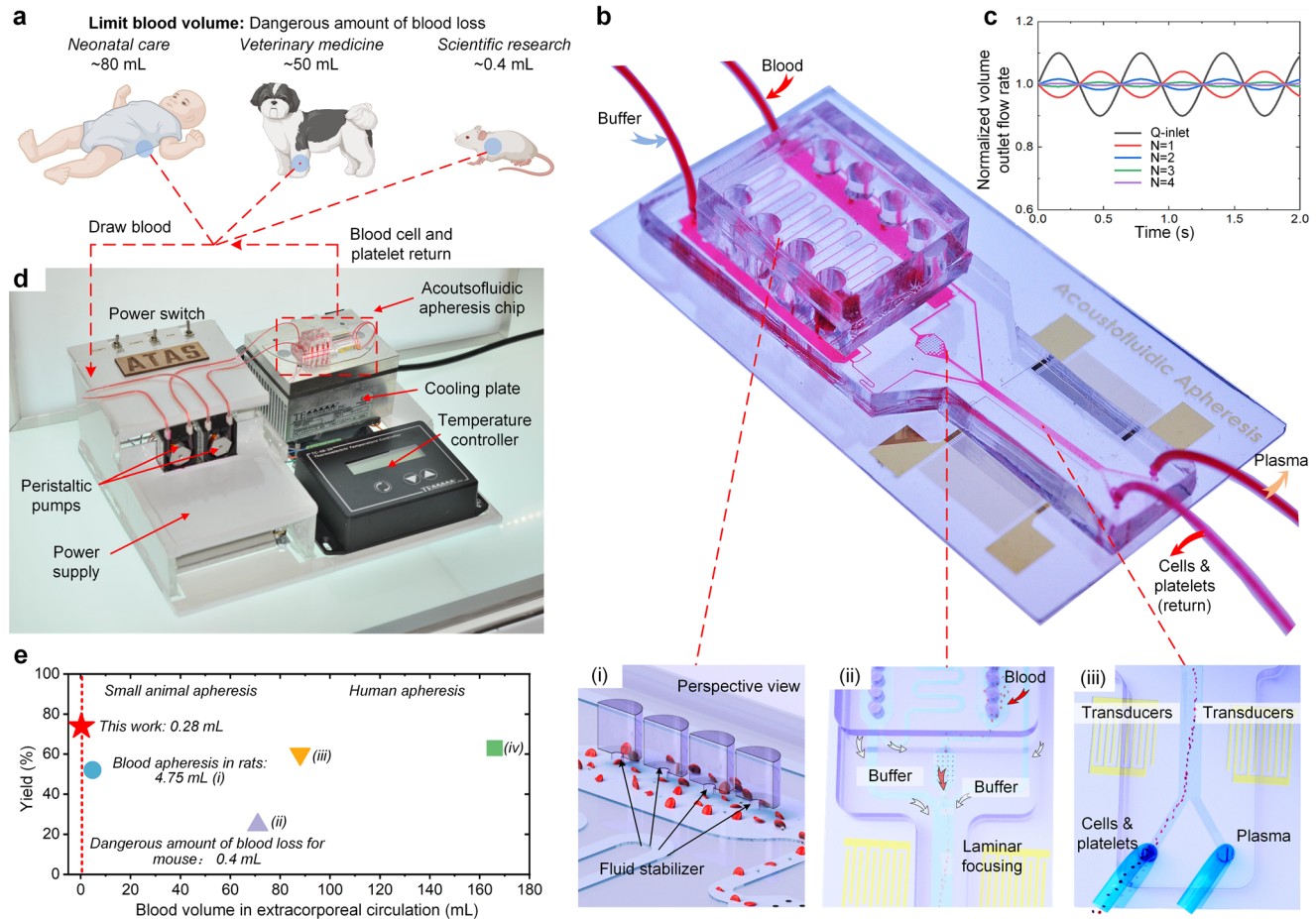

**Fig. 1 | Strategy and prototype of the ATAS. a** The ATAS strategy aims to resolve the obstacles of apheresis in clinical and laboratory practices such as neonatal care, veterinary medicine, and scientific research. **b** The ATAS strategy uses an acoustofluidic microchip where blood cells, platelets, and plasma are separated in a continuous and stable manner. **c** Numerical simulation of fluid stabilizing performance. *N* represents the number of stabilizers. **d** The acoustofluidic therapeutic apheresis prototype. **e** Comparison map of the ATAS with other apparatus reported in the literature: (i) ref. 17, (ii) ref. 44, (iii) ref. 45, and (iv) ref. 46. Figure **a** created with BioRender.com.

blood components is carried out (inset (iii) in Fig. 1b). Acoustic pressure nodes are generated through the interference of two sets of surface acoustic waves produced by opposing pairs of interdigitated transducers. The acoustic radiation force deflects platelets and blood cells toward the acoustic pressure nodes, relocating them into the buffer. This process separates plasma and cellular components in real-time, resulting in an uninterrupted blood component separation. A detailed explanation of the acoustics-based separation mechanism is illustrated in Supplementary Fig. 1. Figure 1c displays simulations of how fluid stabilization responses vary with the number of fluid stabilizers. Notably, using four stabilizers reduces flow rate fluctuations to a negligible level (-0.04% of input fluctuations).

Figure 1d shows the ATAS prototype, which integrates all components into a compact desktop system of 28.0 cm by 22.0 cm. Blood and buffer flows are driven by two peristaltic pumps connected to the microchip via polystyrene tubing with an inner diameter of 0.28 mm. The system also features a temperature controller and a cooling plate to keep the acoustofluidic chip at the optimal temperature, preserving blood integrity during extracorporeal circulation. Due to the small size of the microfluidic channels and interconnecting tubing, the entire system boasts an extracorporeal volume of just 280 μL.

Figure 1e contrasts the ATAS with other apheresis systems for animals found in the literature, including those by Ma et al. [17], Francy et al. [44], Walton et al. [45], and Ahmed et al. [46]. The ATAS stands out as the only system with an extracorporeal volume well below the dangerous

blood loss threshold for mice (approximately 0.4 mL). Furthermore, the ATAS demonstrates superior yield and biocompatibility compared to other systems.

## Constructing stable extracorporeal circulation and achieving highly efficient separation

The ATAS achieves precise, stable extracorporeal blood circulation and highly efficient apheresis separation, as shown in Fig. 2. Key components of the acoustofluidic chip, namely the fluidic stabilizer and acoustic-based separation module, are characterized. Peristaltic pumps ensure uninterrupted and automated extracorporeal blood circulation. Fluid stabilizers have been developed and integrated into the circulation to smooth out the unstable driving force induced by the peristaltic pumps.

A fluidic stabilizer consists of a cavity connected to the main fluid microchannel, as depicted in the three-dimensional structure shown in Fig. 2a. The operational mechanism and numerical results of the cavity-based fluid stabilizer are detailed in Fig. 2b. Initially, the cavity is devoid of fluid and serves as a buffer zone for incoming flux fluctuations. If the incoming flux increases, the cavity compresses the air, creating additional space for the required fluid. Conversely, when the flux decreases, the corresponding fluid is expelled from the cavity to free up space in the microfluidic channels. Importantly, as fluid flux and pressure rise, the liquid level increases while air is compressed. Conversely, air decompresses as the liquid level drops, due to the decrease in fluid flux

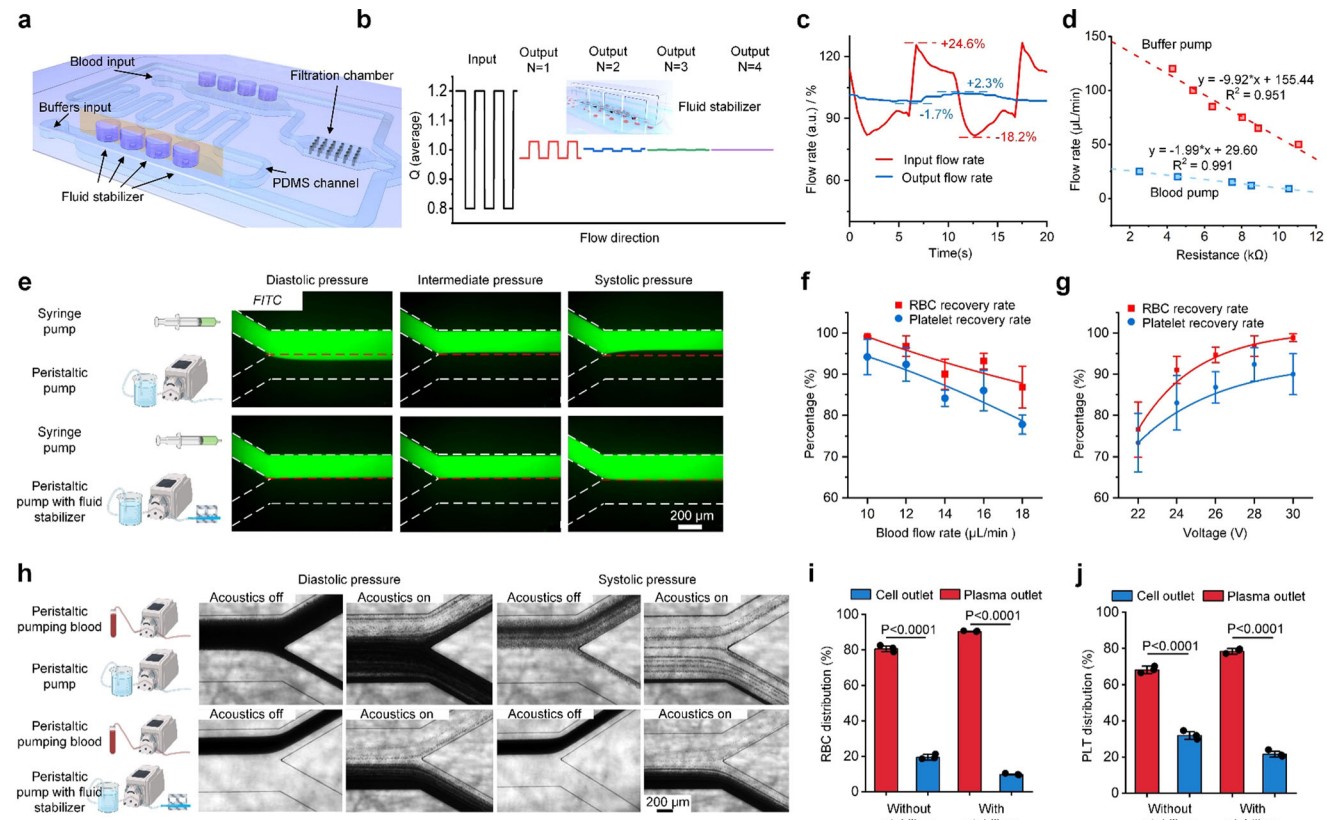

**Fig. 2 | The ATAS provides precise and stable extracorporeal circulation and highly efficient separation for apheresis. a** Schematic view showing the structure of the fluid stabilizer used in the ATAS to generate a steady flow rate. **b** Mechanism and simulation of the fluid stabilizer show that the input flow's fluctuation amplitude is decreased via passing by the cavities connected beside the main channel. **c** Experimental results show the comparison of the input flow rate and output flow rate of the fluid stabilizer as a function of time. **d** The calibration data of flow rate adjustment of the peristaltic pumps used in the ATAS. **e** Comparison of the ATAS and commercial syringe pump regarding flow stability by injecting fluorescent and

non-fluorescent water into the Y-shape channel. Characterization of the red blood cell (RBC) and platelet recovery rate after ATAS processing under varied **f** flow rates and **g** input voltages. Data represent the mean ± SD; $n$ = 3 (flow rates) and 3 (input voltages). **h** Microscopy images show the separation of cellular components from plasma using the ATAS when a fluid stabilizer is not equipped or equipped. Distribution of **i** RBCs and **j** platelets in the samples collected from the cell and plasma outlets. Data represent the mean ± SD; $n$ = 3 (RBCs) and 3 (platelets). The statistical analysis was performed using a two-sided Student's $t$-test. Data are graphed as the mean ± SD. Source data are provided as a Source Data file.

and pressure. Consequently, regardless of the dramatic fluctuations in systolic or diastolic pressures, the fluid controller maintains a relatively stable output pressure, resulting in stabilized flow rates. Our simulation result (Supplementary Figs. 2–4) demonstrates that when the input flow exhibits square-wave-like fluctuations with an amplitude of 20%, the fluctuation can be reduced to <5% after passing through a single stabilizer, and flow rates become stable (fluctuation < 0.04%) after passing through four stabilizers. The integration of the four cavities is shown in Supplementary Fig. 5. Additionally, an on-chip filter has been designed to trap clots or bubbles, as shown in Supplementary Fig. 6.

We quantitatively analyzed flow rates before and after implementing the fluidic stabilizer, illustrating the changes over time in Fig. 2c. Initially, flow rates showed fluctuations of around ±20%. After passing through the fluid stabilizer, these fluctuations were minimized to <±2.3%. By adjusting the resistance of the peristaltic pump, we could precisely control flow rates within the system. Figure 2d demonstrates how we calibrated the flow rates of the two peristaltic pumps used for circulating buffer or blood.

Figure 2e compares the stability of the fluid in the ATAS circulation with that of a commercial syringe pump. A fluorescent fluid is ejected from a syringe pump towards an inlet of a Y-shaped channel. Simultaneously, the ATAS system introduces a non-fluorescent fluid into the second inlet of the Y-shaped channel. Without the use of fluid stabilizers, the interface between the two fluids experiences significant

fluctuations due to the instability of the flux from the peristaltic pumps. However, once the fluid stabilizer is implemented, the ATAS maintains a stable flow rate with minimal fluctuations, resulting in a constant interface between the fluorescent and non-fluorescent water.

We also characterized the ATAS's acoustic-based separation module by evaluating the effectiveness of its blood separation. In this module, acoustic radiation forces facilitate the transportation of objects such as blood cells or platelets and separate them based on size and other physical properties. Cellular components are separated from the blood and transferred into the sheath fluid, while antibodies remain in the plasma component due to their solubility. Supplementary Fig. 7 displays photographs and microscope images of the entire channel, including the inlet/outlet regions, when acoustic waves are activated.

We analyzed red blood cell (RBC) and platelet recovery rates under different conditions to optimize the blood separation. We introduced whole blood from mice into our system and measured the quantities of blood components in the cell-platelet mix and plasma using a commercial blood analyzer, as shown in Fig. 2f. Initially, we set the input voltage to 28 V, used a 5% dextrose solution for the sheath flow, and maintained a sample-to-sheath flow rate ratio of 1:6. This setup achieved recovery rates of over 95% for RBCs and over 90% for platelets. The ATAS system can process an experimental mouse's entire blood volume in less than two hours.

Figure 2g shows the recovery rate as a function of input voltage. The input voltage for the electric signal is increased at increments of 2 V from 22 to 30 V, while the separator throughput is set at 12 µL min$^{-1}$. An observable trend emerges, directly correlating increased input voltage with higher recovery rates. The recovery rates of RBC and platelet increased from approximately 75% at an initial voltage of 22 V to approximately 95% and 90%, respectively, at an input voltage of 30 V. Furthermore, we quantified recovery rates when altering the sample/sheath flow ratio and sheath fluids, as depicted in Supplementary Fig. 8. It is worth noting that increasing the sheath flow from 60 to 120 µL min$^{-1}$ maintains RBC and platelet recovery rates of ~95% and ~90% respectively. The sheath fluid was replaced with plasma to preserve blood volume after separation. We evaluated the effects of utilizing different sheath fluids, such as 5% albumin, 5% dextrose, PBS, 10% dextrose, and 0.9% saline solutions. We found the choice of fluid had a negligible effect on the recovery rates for RBCs and platelets. This result demonstrates the flexibility of our ATAS system, as it does not constrain the fluids' properties and thus allows for various fluid applications, such as intravenous fluids, to return blood cells.

A stable flow rate is essential for smooth blood circulation and efficient separation in therapeutic apheresis. We evaluated the working conditions of the therapeutic apheresis system and compared its performance with and without using the fluid stabilizer in the ATAS. Figure 2h illustrates the blood flow and separation situation at the outlet region of the acoustofluidic chip when two peristaltic pumps drive the system without the assistance of a fluid stabilizer. Without acoustic waves, blood and buffer flow rates fluctuate, causing significant changes in blood cell concentration. The inconsistent flow rate disrupts the separation process when the acoustic waves are activated. This disruption results in some blood cells exiting through the plasma outlet and decreasing the recovery rate.

In comparison, the performance significantly improves when incorporating the fluidic stabilizer. Blood cells are directed to their respective outlets when the acoustics are turned off and on. Figure 2i and j show the distribution of RBCs and platelets in the cell outlet and plasma outlet after acoustofluidic separation. When the stabilizer is not used, around 80% of the RBCs and 70% of the platelets are found in the cell outlet, indicating that ~20% of the RBCs and 30% of the platelets end up in the plasma outlet and are lost. With the assistance of the stabilizer, the recovery rates of RBCs and platelets can be increased to approximately 90% and 80%, respectively. In summary, the fluid controller ensures smooth and stable flow in the extracorporeal circulation, thereby improving the efficiency of acoustofluidic-based therapeutic apheresis.

## Characterization of ATAS using mouse models

To ensure a comprehensive characterization of the ATAS in terms of its blood component fractionating abilities, we conducted extensive testing on the separated blood components, with the results shown in Fig. 3. The experimental and characterization procedure is outlined in Fig. 3a. After ATAS processes 1 mL of blood from an experimental animal, both outlet samples are collected for evaluation. The soluble components remaining in the plasma are evaluated via ELISA, while the blood cells and platelets are identified using a blood analyzer and flow cytometry.

The distribution of RBCs, platelets, and IgG (as a representative antibody in plasma) in the two outlets are compared to confirm the separation efficiency. With acoustics off, RBCs, platelets, and IgG all exit through the plasma outlet (Fig. 3b). With acoustics on, over 90% of RBCs and platelets shift to the cell outlet, with nearly 80% of IgG staying in the plasma outlet. These findings demonstrate the effective separation of cellular blood components from plasma soluble with high efficiency and recovery.

In addition, we examined the functionality of the recovered RBCs and platelets following acoustofluidic separation. We tested and compared levels of P-selectin, which serves as an indicator of platelet function. The size of mean platelet aggregates directly correlates with the expression of P-selectin, indicating that increased levels of P-selectin reflect platelet activation[47,48]. Figure 3c shows no significant difference in P-selectin levels post-acoustofluidic separation compared to the initial blood sample.

The concentration of hemoglobin, which is the component of RBCs responsible for oxygen transport, in the sample collected from the cell outlet and plasma outlet, is also compared (Fig. 3d). Under ideal parameters, i.e., an input voltage of 30 V and a blood flow rate of 10 µL min$^{-1}$, only ~1% of hemoglobin is lost when blood cells are transported into the buffer. In addition, the morphology and condition of blood components were characterized using Wright staining and scanning electron microscopy (SEM), as shown in Supplementary Fig. 9. The characteristics of both platelets and RBCs align with general hematology guidelines. The platelets appear as small violet and purple granules, and the RBCs exhibit a circular shape with a transparent center. SEM images demonstrate that the RBCs are uniform in diameter with a biconcave shape, consistent with the appropriate morphology for RBCs. In summary, the results indicate that ATAS is a gentle, biocompatible procedure that preserves the characteristics of blood components while delivering high efficiency and recovery in terms of separation performance.

Figure 3e and f present detailed flow cytometry profiles of the cell and plasma outlet. These profiles, based on forward scattering and side scattering levels, allow for the distinction of blood cells and platelets. A higher presence of blood cells and platelets is evident in the blood cell fraction compared to the plasma fraction. A quantitative comparison of the number of blood cells and platelets reveals a removal rate exceeding 96% for both RBCs and platelets. Figure 3g and h demonstrate the labeling of the separated samples with anti-CD61 antibodies, which are the markers for platelets. After separation by the ATAS, the count of anti-CD61 positive events in the cell outlet is about three times higher than that in the plasma outlet.

To verify the ATAS system's capability to separate immunoglobulins, we added monoclonal antibodies to non-reactive whole blood, mimicking highly sensitized patient conditions. We then used flow cytometry to measure the levels of monoclonal antibodies after separation. Prior to employing acoustofluidic separation, we ensured that unbound antibodies were unaffected by our microchip and acoustic waves and did not mix into other fractions, as shown in Supplementary Fig. 10. We introduced anti-human CD3 monoclonal antibodies to mice whole blood and processed it with the ATAS system. In the ATAS system, cellular components were separated from the plasma and transferred to a replacement fluid, i.e., saline, to maintain blood volume. The plasma portion, as well as the immunoglobulins and other solvable antibodies, was not transferred. Thus, after processing by the ATAS system, the cellular components would not contain immunoglobulins. Post-separation, these antibodies were assessed on human cells targeted by the antibody. Figure 3i shows a marked difference in expression between samples from the cell outlet and those from the plasma outlet. As Fig. 3j indicates, compared to the PBS negative control, the mean fluorescence intensity at the cell outlet is significantly lower than at the plasma outlet.

We also tested whether the device could separate anti-MAMU A01 antibodies. MAMU A01 is a prevalent major histocompatibility complex known to correlate with simian-human immunodeficiency virus infection[49]. We added anti-MAMU A01 antibodies to blood from non-MAMU A01 rhesus monkeys and processed the blood through our system. As shown in Supplementary Fig. 11, the results indicate that anti-MAMU A01 antibodies did not bind to any cells in the whole blood of the non-A01 monkey. These unbound antibodies were separated into the plasma outlet, and there was no evidence of anti-MAMU A01 antibodies in the cell outlet. Collectively, these results

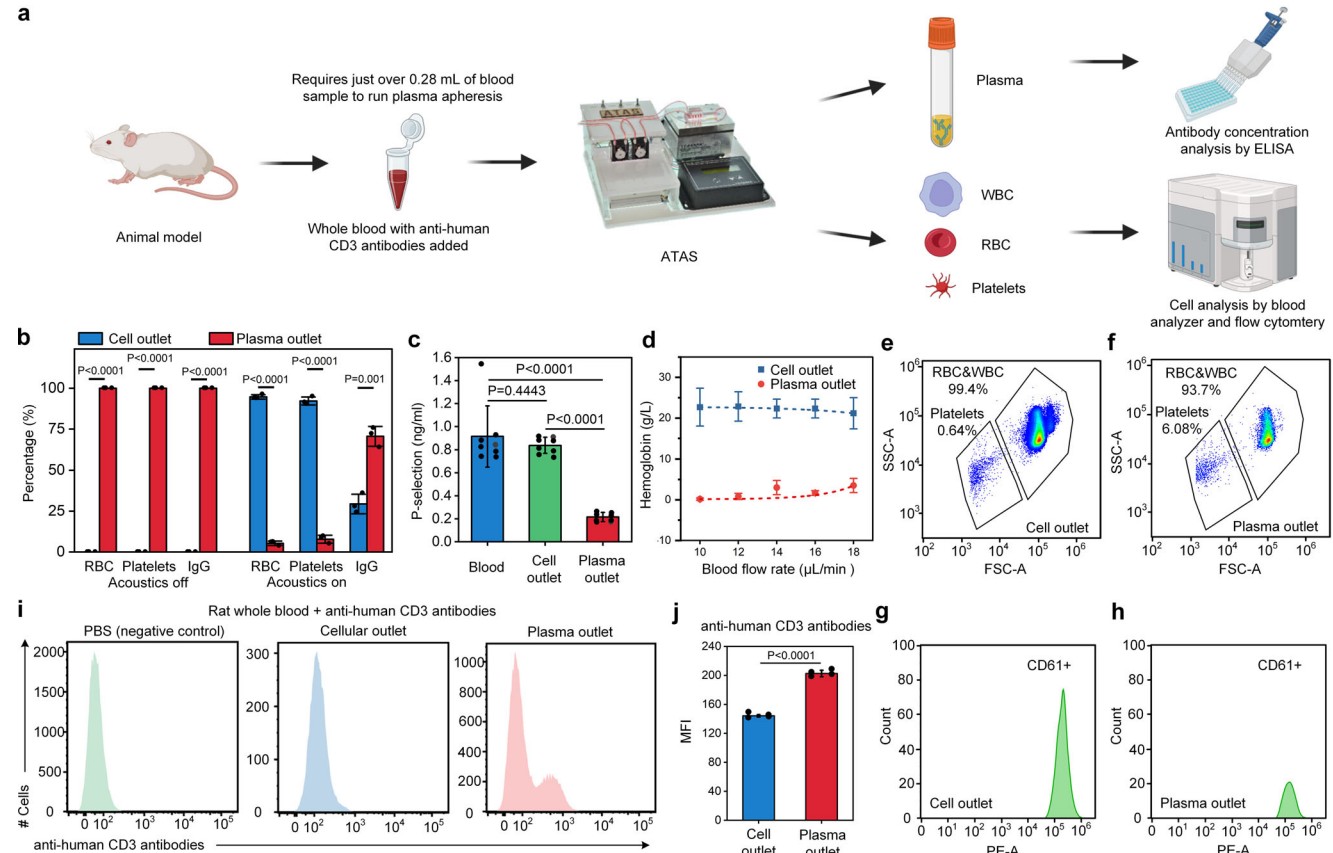

**Fig. 3 | High efficiency and low dead volume enable the use of the ATAS on mouse models. a** Schematic view showing the process of plasma apheresis and performance characterization. **b** Distribution of cellular components (e.g., RBCs and platelets) and soluble components (e.g., IgG) in the sample collected from the ATAS's cell outlet and plasma outlet when acoustics are off vs on. Data represent the mean ± SD; *n* = 3. **c** The level of P-selectin in the whole blood sample was collected from the cell outlet and plasma outlet. Data represent the mean ± SD; *n* = 8. **d** The hemoglobin concentration in the sample was collected from the cell outlet and plasma outlet. Data represent the mean ± SD; *n* = 3. Flow cytometry data showing the number of blood cells and platelets in the sample collected from **e** cell outlet (left panel) and **f** plasma outlet of the ATAS. Count of platelets (identified by CD61+) in the sample collected from **g** cell outlet and **h** plasma outlet of the ATAS. **i** Isolation of unbound monoclonal antibodies against human CD3 from mice whole blood. **j** Comparison of the levels of anti-human CD3 monoclonal antibodies present in plasma and cell outlet. Data represent the mean ± SD; *n* = 4. The statistical analysis was performed using a two-sided Student's *t*-test and one-way ANOVA with Tukey's post-hoc test. Data are graphed as the mean ± SD. Source data are provided as a Source Data file. Figure **a** created with BioRender.com.

confirm the capability of the ATAS to successfully separate unbound antibodies present in blood from other cells, thereby performing plasmapheresis.

## Therapeutic apheresis procedure for desensitization in sensitized animals

Utilizing ATAS, we can now conduct in-vivo procedures on mouse models to remove donor-specific antibodies that are detrimental to the transplanted organ in a blood-saving, highly efficient, and automated manner. This approach has broad applicability in treating a variety of antibody-mediated diseases. We conducted the first therapeutic apheresis procedure using the ATAS, as depicted in Fig. 4. In this procedure, the recipient animals (C57BL/6 mice) were sensitized with BLAB/c donor skin transplantation. The sensitized recipient mice typically develop an allogeneic response after donor skin transplantation. Consequently, donor-specific antibodies accumulate in the blood, leading to antibody-mediated rejection after organ transplantation. To combat this, we used ATAS for therapeutic apheresis in recipient mice to reduce preformed donor-specific antibodies (Fig. 4a), mirroring the situation in sensitized human recipients awaiting organ transplants. As shown in Fig. 4b, the experimental setup connected the ATAS to the animal model via vein catheters. As a proof of concept, blood is returned in various ways. In the case of testing the separation performance for donor-specific antibodies, only one

catheter is connected for blood withdrawal from the mouse. Cell components are returned with PBS (~500 μL) after the procedure. For therapeutic apheresis purposes, both a withdrawing catheter and a returning catheter are necessary to return extracorporeal circulation to the animal instantly.

The levels of donor-specific antibodies in the samples collected from the cell outlet and plasma outlet were assessed using flow cytometric crossmatch, as depicted in Fig. 4e. Subsequently, we monitored the donor-specific antibody level in blood samples of the recipient mice via cheek bleeding. Figure 4d and e compare the donor-specific antibody levels in the blood of the recipients before and after the therapeutic apheresis procedure. At the time of the procedure, all graft recipients displayed significantly elevated levels of donor-specific antibodies. ~500 μL of circulating blood is processed through the device *via* the inferior vena cava, achieving a 10–12 μL/min throughput. The serum donor-specific antibody levels in the recipients were significantly reduced after the procedure, decreasing by approximately 40% and 56% when measured using B-cell crossmatch and T-cell crossmatch, respectively. These results demonstrate that exchanging plasma is an effective approach for mouse models, like its use in desensitization in sensitized human patients. In summary, these findings prove that our ATAS system can perform therapeutic apheresis for small animals with potential translation to neonates and small children.

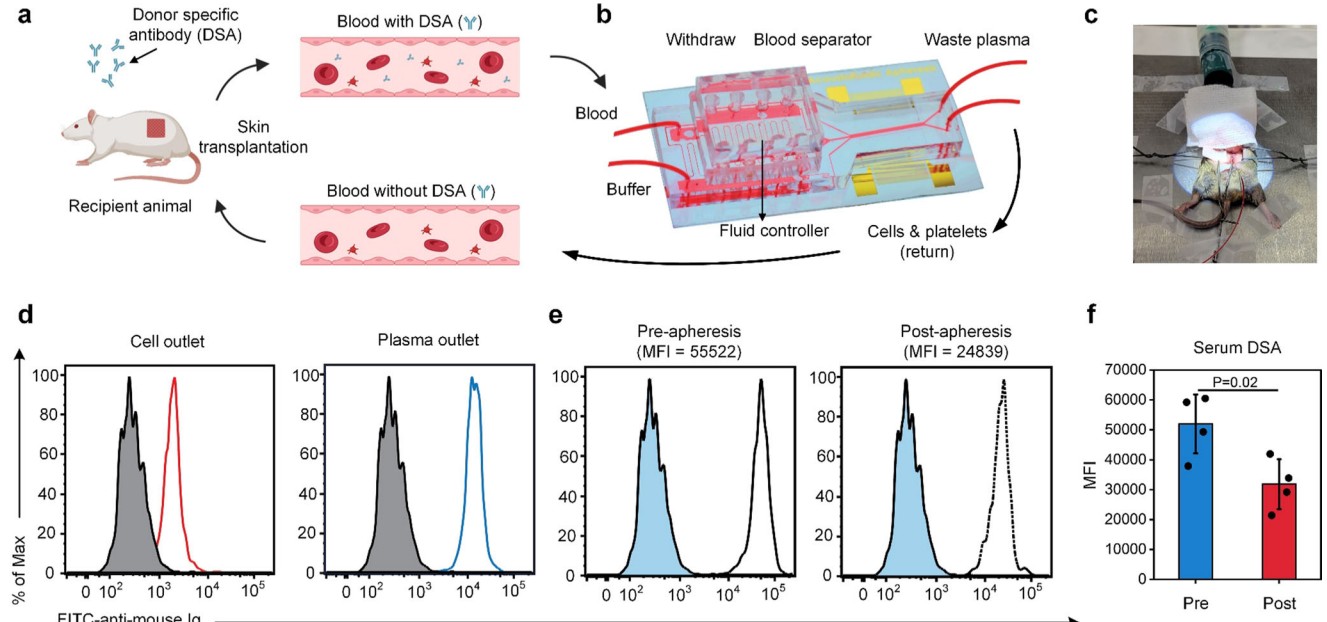

**Fig. 4 | Therapeutic apheresis for a sensitized mouse recipient to treat transplant rejection via the ATAS. a** and **b** Schematic of treatment procedure for allogeneic skin graft transplanted mouse recipient by reducing the donor-specific antibody (DSA) level in the recipient's blood to reduce transplant rejection. **c** Photographs of the tubing of the ATAS and the connection via vein catheters. **d** DSA level testing for the sample collected from the plasma outlet and cell outlet of the ATAS. **e** and **f** Serum DSA level testing for the recipient animals' pre- and post-apheresis. Data represent the mean ± SD; *n* = 4. The statistical analysis was performed using a two-sided Student's *t*-test. Data are graphed as the mean ± SD. Source data are provided as a Source Data file. Figure **a**, **b** created with BioRender.com.

## Discussion

As an in vivo proof of concept, we successfully applied our acoustic-based plasmapheresis system in a sensitized mouse model, primarily targeting unbound pathological antibodies, namely donor-specific antibodies. Donor-specific antibodies, comprising both preformed and de novo types, are a growing challenge in organ transplantation[50,51]. Patients who require organ transplantation often develop anti-HLA antibodies prior to their transplantation, whether through exposures via blood transfusion, pregnancy, or organ transplantation[52,53]. Following sensitization, these patients are less likely to find a suitable organ and have a higher chance of rejecting the transplanted organ, given their heightened immunologic risks. For this reason, these sensitized patients are often on waitlists for extended periods, accounting for approximately 30% of the waitlisted patients[54,55]. Strategies to decrease alloantibodies, the antibodies formed in response to transplanted organs, through desensitization are thus an appealing approach as they target a broad patient population and have been associated with a substantial survival benefit[56].

To date, desensitization strategies in transplantation have been almost exclusively based on antibody removal via plasmapheresis. However, the scope of most existing apheresis instruments is limited, as they cater exclusively to clinically stable adults. Given that many new therapeutics originate from mouse or non-human primate models, the need for suitable apheresis devices for these models hinders the translation of innovative strategies into clinical practice, particularly for smaller patients like infants and children. This study successfully demonstrates plasmapheresis in a small animal model. Besides, our device, through modifications to the chip design and working conditions, has the potential to separate many types of blood components, such as red blood cells, platelets, lipoproteins, viruses, and extracellular vesicles. Although some challenges remain in realizing these functions, the ATAS platform can pave the path to various of biomedical applications.

Even with successful plasmapheresis in an in vivo sensitized mouse model, we found several limitations to the ATAS method. First, we observed incomplete separation of antibodies and cells (Fig. 3). We will address this shortcoming in future prototypes with improvements in the design of the acoustic transducer, which will result in greater acoustic radiation forces and, hence, stronger separating power. Additionally, the fluctuation due to pulsatile flow can be further reduced by increasing the number of cavities in the fluid stabilizer. We are also exploring the possibility of incorporating three-dimensional standing acoustic waves generated in hybrid channels reported in our previous work[23]. This design has an acoustic intensity across the microfluidic channel that is several times larger than the current device design, allowing an increased throughput of 125 µL min⁻¹. This will significantly shorten the processing time by around tenfold. Finally, the effects of repeated plasmapheresis on the host with our device are currently unknown and warrant further investigation. The safety threshold values (such as Hgb level, platelet count, etc.) must be investigated and verified for a subject to perform therapeutic apheresis on the ATAS platform.

Mouse models are widely used in biomedical research, but their small size often limits the compatibility of existing technologies. As a result, researchers often need to develop their own methods for experimentation. This results in substantial heterogeneity and variance in methodology from researcher to researcher within the same field. Such discrepancies in experimental techniques hinder the establishment of standards, comparison of results, and translation of therapies from animals to humans. Our ATAS system provides researchers with a possible common standard to conduct experiments and reduce variance in methodology.

In summary, ATAS provides a comprehensive and consistent method for conducting blood-related experiments on animal models. This will significantly advance medical research that can be used to treat critically ill patients, including adults, neonates, infants, and children. By ensuring standardization, we can translate the results of

these studies to the bedside, paving pathways for translational research to impact those who need it most. The ATAS enables innovative solutions to previously unattainable challenges in preclinical and clinical settings.

## Methods

### Manufacturing the ATAS chip

The ATAS chip described in the Supplementary Information is comprised of two main components: a piezoelectric substrate and a multilayer polydimethylsiloxane (PDMS) microfluidic channel. Electrodes on the piezoelectric substrate are fabricated through e-beam evaporation, photolithography, and a lift-off process. The spacing of the interdigitated electrode is 50 μm, the length of the electrode is 10.5 mm, and the distance between two pairs of interdigitated electrodes is 9 mm. The PDMS microfluidic channel is produced by stacking three PDMS sheets on each other. The first and second sheets contain microfluidic channels and through-layer cavities. The channels are fabricated using soft lithography, involving these key steps:

1. A layer of SU8 photoresist is applied to a silicon wafer via spin-coating and UV-lithography, creating a mold with the desired channel pattern.
2. A mixture of a curing agent (Dow Corning, USA) and PDMS base in a 1:10 weight ratio is spread over the SU8 mold. After baking at 65 °C for an hour, the PDMS layer solidifies.
3. Inlet and outlet holes are added using a punch (EMS, 69039-10).
4. The three PDMS sheets are bonded following plasma treatment, ensuring a secure seal between the layers.

The microfluidic channel for blood component separation has a width of 400 μm and a height of 100 μm. Subsequently, the piezoelectric substrate and PDMS subcomponents undergo an oxygen plasma treatment before bonding. Finally, the device is baked at 90 °C overnight to create a robust bond between the components. This allows the ATAS chip to generate acoustic waves and perform blood component separation within its microfluidic channels.

### Configuring the ATAS

Our ATAS has various components, including a cooling plate, a power supply, peristaltic pumps, a temperature controller, and other electronic accessories. Specifically, we used the following equipment. Model P625 pumps (Instech Laboratories, Inc., Plymouth Meeting, Pennsylvania, USA) transported liquids to the corresponding chip inlets via plastic tubes (Smith Medical International, USA). The acoustofluidic microchip was placed on a cold plate cooler (Model: CP-031) controlled by a temperature controller (Model: TC-48-20 PWM). Both components were sourced from TE Technology Inc (Traverse City, MI, USA). The cooling plate was configured to maintain a temperature of 10 °C. All the components were powered by a 12 V supply, provided by a DC power supply (Model: PS-12-8.4A), also sourced from TE Technology Inc. During the experiments, the acoustofluidic microchip was actuated by radio frequency electric signals produced using an amplifier (100A250A, Amplifier Research, USA) and a function generator (E4422B, Agilent, USA). The frequency of acoustic waves is 20 MHz. The input power was evaluated using an oscilloscope (DPO4104, Tektronix, USA).

### Recording and micrographing

An upright microscope (BX51WI, Olympus, Japan) and a CCD camera (CoolSNAP HQ2, Photometrics, USA) were used to document the separation performance. The device was secured to the microscope stage, and the CCD captured the entirety of the separation procedure. The resulting film was processed and studied using Image J (NIH, USA).

To gain insights into the pumping performance of the prototype, we compared it with commercialized syringe pumps. The syringe pumps used for this purpose were the neMESYS model from Cetoni

GmbH, Germany. To distinguish between the two types of fluids used in the comparative analysis, we introduced the fluorescent dye Fluorinert FC-40 (Sigma-Aldrich, Missouri, USA) into the fluid driven by the syringe pump. This step allowed us to differentiate and assess the performance of both our ATAS prototype and the standard syringe pump.

### Human whole blood

We sourced frozen plasma samples and fresh human whole blood from Zen-Bio, Inc. (North Carolina, USA). Human whole blood samples were collected using microcentrifuge tubes. They were subjected to examination utilizing a blood analyzer (COULTER Ac·T diff2, Beckman Coulter Inc., California, US) and a flow cytometer (BD FACSCanto II). The IgG levels in mouse blood were quantified using a Mice IgG ELISA kit (88-50400-22, Invitrogen).

Throughout the experiments, both the original blood samples and the processed samples were collected to determine their respective volumes. The separation efficiency and recovery rates of red blood cells (RBC), platelets, and IgG were subsequently calculated based on the results of the whole blood analysis and the sample volumes.

### Ex vivo and in vivo plasmapheresis

We collected blood samples from mice, rats, and nonhuman primates (NHP) for use in acoustofluidic apheresis and to isolate peripheral blood mononuclear cells (PBMC). PBMCs were obtained through density-gradient centrifugation, using 90% separation media sourced from Sigma. For ex vivo antibody separation, APC anti-human CD3 antibody (Clone: SP34-2, BD Biosciences) or PE anti-MAMUA01 antibody (Clone: P12, Nonhuman Primate Reagent Resource, RRID: AB_2819290) were added to 200 μL of PBS or whole blood. In the case of in vivo acoustofluidic apheresis, the procedures were carried out on mice under isoflurane anesthesia (5% for induction and 2% for maintenance). Following either ex vivo or in vivo acoustofluidic apheresis (from either cell or plasma outlets), the collected samples were centrifuged at 3000 rpm, and the supernatant was extracted for further analysis.

For donor-specific antibody detection, we employed the previously described flow crossmatch utilizing cells with specific antibody specificities. The samples were then briefly incubated with target cells, and IgG DSA was measured through a flow cytometric crossmatch conducted on a BD LSRFortessa™ (BD Biosciences, San Jose, CA, USA). The data was analyzed using FlowJo software version 10 (Tree Star, Ashland, OR, USA). Statistical analyses were conducted using GraphPad Prism software version 9.0 (GraphPad Software, San Diego, CA, USA). Statistical comparisons between different samples were performed using the paired Student $t$-test with an alpha level of 0.05. All mice were used and cared for following the guidelines and compliance set forth by the Duke Institutional Animal Research Ethics Committee.

### Wright staining and SEM

Blood samples were deposited onto Shandon™ Double Cytoslides™ (Thermal Fisher Scientific, USA) to prepare blood smears for observation. A glass slide was then positioned over the samples to create blood smears. Subsequently, the samples were stained with *Stain*RITE® Wright Stain Solution, procured from Polysciences Inc. (PA, USA). The staining procedure followed the guidelines outlined in the stain solution's manual.

To perform blood cell morphology observation via SEM, the blood samples were placed on the Shandon™ Double Cytoslides™. To secure the positioning of the cells, the slide was immersed in a 4% paraformaldehyde solution. Following this fixation step, a thin layer of gold film was applied to the specimen. Finally, the sample was observed under SEM, and images were taken using computer software.

## Reporting summary

Further information on research design is available in the Nature Portfolio Reporting Summary linked to this article.

## Data availability

All data supporting the findings of this study are available within the article and its supplementary files. Any additional requests for information can be directed to the corresponding author (T.J.H., email: tony.huang@duke.edu), and will be fulfilled by the corresponding author. Source data are provided with this paper.

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

## Acknowledgements

We acknowledge support from the National Institutes of Health (R44HL140800, U19AI131471, R01HD103727, and R01GM132603).

## Author contributions

M.W. and Z.T.M. developed the system concept and led the experimental work. Z.T.M. and B.L. conducted experimental work and figure drawing. X.C.X. conducted numerical simulations. J.K., L.P.L., and T.J.H. guided the experimental design, figure drawing, and paper writing. M.W., Z.T.M., Y.G., Z.H.M., and J.P.X. conducted the experiments and data analyses. J.H.Y. and N.U. contributed to biological sample preparation and process. I.J.A., S.J.K., E.T.C., J.K., L.P.L., and T.J.H. provided guidance and contributed to the design and analysis throughout the project.

## Competing interests

T.J.H. has co-founded a startup company, Ascent Bio-Nano Technologies Inc., to commercialize technologies involving acoustofluidics and acoustic tweezers. All other authors declare that they have no competing interests.
