## [Peer Review File · Nature Communications]

REVIEWER COMMENTS

Reviewer #1 (Remarks to the Author):

This manuscript describes an acoustofluidic-based therapeutic apheresis (ATAS) system that is designed for blood component separation in subjects with small total blood volumes such as small animals and neonates. The need for such a device/instrument is sorely needed and will have a significant impact on patient care when commercialized. The manuscript is well written; however, it was difficult to visualize how the blood flows through the device and how the components were being separated by the device. I think the diagram of the device could be improved. Perhaps have a single figure just describing the blood flow pattern and how the blood cells are being separated from the plasma would be useful for the readers. The current figure is not detailed enough.

Abstract:

I suggest revising the first sentence to this: Therapeutic apheresis primarily aims to selectively remove pathogenic substances such as antibodies that trigger various symptoms and diseases.

Introduction:

1. What is the maximum volume the system can handle? This is important to include. Also, how long does it take to process blood?
2. Last paragraph of introduction: I suggest revising "limited blood volumes" in the first sentence to "small blood volumes".

Results:

1. What is the anticoagulant used in the system?
2. Are the antibodies removed simply as a result of the plasma being removed or does the device selectively pull out the antibodies from the whole blood? When plasma is removed, is a replacement fluid given to maintain blood volume?
3. What is the fluid balance of the subject after the procedure? Positive? Negative? Neutral?

4. What types of adverse effects can be expected/extrapolated when used in humans based on experience with mice or other small animals?
5. If this device were to be used as a method for red cell exchange, could non-diseased red blood cells be infused as a replacement?
6. What are the threshold values for lab parameters if applicable (eg hgb/hct, platelet count, etc) for it to be safe to put a subject on this device?
7. Is there a maximum cell count or antibody titer for this device to be effective? If yes, what are they? I am wondering if the system can get "saturated". Does efficiency vary according the cell count? If so, what is the relationship between efficiency and lab parameters?

Discussion

1. An explanation of how acoustics are able to separate cells from plasma would allow readers who have limited or no background in this field to appreciate the concept/idea more and make the manuscript more understandable.
2. Discuss potential adverse effects on cells that acoustics have. Will it cause hemolysis which has a lot of downstream adverse consequences?

Reviewer #2 (Remarks to the Author):

The manuscript describes therapeutic apheresis, and particularly focuses upon apheresis for small blood volumes in animal use, an interesting justification for acoustofluidics methods in this well-established technique. Overall, the work is sound and I can certainly see this contribution appearing in Nature Communications with its rather complete approach and compelling results, following all the way through to animal model use in relation to identifying and separating pathological antibodies in blood. Figure 4 makes it most clear, perhaps, that this method is a very interesting and powerful technique.

I would encourage the authors to expand slightly on the idea of apheresis in mice, given the penchant the NIH has in funding and using murine models for nearly all types of scientific research. There are

many scientific studies underway that rely on the mouse model as a consequence, arguably more than there should be given the limitations of mice in representing human health, but nonetheless mice are the dominant animal model in research. This contribution is especially useful in that context. Citation to a few acoustofluidics works in blood separation seems appropriate as well; for example, Zhang, et al. LoC 21:904-15 2021.

The frequency of the acoustic device appears to be missing, as does the rationale for the channel width and height versus attenuation length of the SAW in the substrate or perhaps the wavelength of the SAW and in the fluid. These aspects are important for the reader to understand the particle manipulation mechanism, at least as it is intended to be used by the authors in the work as reported.

The introduction of the compressible air cavities is a nice solution to an annoying problem when using pulsatile flow pumps. The traditional solution is of course syringe pumps, yet they lack the ability to run for long periods and continually provide fluid. Noting the statement on p8, "With the assistance of the stabilizer, the recovery rates of RBCs and platelets can be increased to approximately 90% and 80%,..." is there anything further that could be done to improve these results? A second matching stage? Some may feel that greater separation fractions are needed in apheresis, and some thoughts on how that might be possible with some simple additions to your effort might be welcomed.

The ordering of the display items in Figure 1 (a-e) is a little odd; I don't know what I'd suggest to do differently, but it breaks the usual canon of left-to-right and top-to-bottom fairly significantly. If it can't be fixed so be it.

p12: "through simple modifications to the chip design and working conditions, has the potential to separate various blood components,..." --- This should really be shown or rewritten a bit, because it is indeed difficult to isolate rare cells or RBCs from WBCs, and the text here gives the impression that any of these actions might be possible.

p4: micron-level -> micro-scale

p5: singular blood stream

Itemized list of response to reviewers' remarks

(Black italic: Editor's remarks; Blue type: Our response; Additions/modifications to the manuscript and Supplementary Materials are highlighted in yellow)

Reviewer #1 (Remarks to the Author):

This manuscript describes an acoustofluidic-based therapeutic apheresis (ATAS) system that is designed for blood component separation in subjects with small total blood volumes such as small animals and neonates. The need for such a device/instrument is sorely needed and will have a significant impact on patient care when commercialized. The manuscript is well written; however, it was difficult to visualize how the blood flows through the device and how the components were being separated by the device. I think the diagram of the device could be improved. Perhaps have a single figure just describing the blood flow pattern and how the blood cells are being separated from the plasma would be useful for the readers. The current figure is not detailed enough.

Our Response:

We appreciate the valuable feedback and time the reviewer devoted to evaluating our manuscript. We have prepared a point-by-point response to the comments and the revised manuscript. The changes have been highlighted in yellow for further review.

Per the reviewer's comments, we have added an extra figure to explain the mechanism of acoustics-based separation. The figure is inserted in Supplementary Fig. 1. The modification is as follows.

In the manuscript's second paragraph of the "Strategy and prototype of the ATAS" section, we added the following sentences:

"Acoustic pressure nodes are generated through the interference of two sets of surface acoustic waves produced by opposing pairs of interdigitated transducers. The acoustic radiation force deflects platelets and blood cells toward the acoustic pressure nodes, relocating them into the buffer. This process separates plasma and cellular

components in real-time, resulting in an uninterrupted blood component separation. A detailed explanation of the acoustics-based separation mechanism is illustrated in Supplementary Fig. S1.”

In the Supplementary Materials, we added the following sentences:

Supplementary Figure 1 | Schematic about the mechanism of acoustics-based separation.

Abstract:

I suggest revising the first sentence to this: Therapeutic apheresis primarily aims to selectively remove pathogenic substances such as antibodies that trigger various symptoms and diseases.

Our Response:

Thank you for your valuable suggestion. We have made the necessary modifications to the introduction based on your comment.

Introduction:

- 1. What is the maximum volume the system can handle? This is important to include. Also, how long does it take to process blood?*

Our Response:

We appreciate the valuable questions that can help us improve our manuscript. Our ATAS platform typically handles a throughput range of $12 \mu\text{L}\cdot\text{min}^{-1}\sim 18 \mu\text{L}\cdot\text{min}^{-1}$. This range suggests that is capable of processing the entire whole blood volume contained by an experimental mouse in ~ 2 hours. Further improvements can be made by using a three-dimensional acoustic standing wave generated in PDMS / glass hybrid channel as seen in our other work.^{R1} This design has an acoustic intensity several times greater across the microfluidic channel than the current design, allowing for an increase in throughput to $125 \mu\text{L}\cdot\text{min}^{-1}$. This will significantly shorten the processing time by around tenfold.

Per the reviewer's comments, we have modified the *Introduction* to include this information and discussed the potential improvements in the *Discussion*.

In the last paragraph of *Introduction*, we added the following sentences:

“Our ATAS permits the separation of unbound and pathogenic donor-specific antibodies from recipient blood with minimal loss of blood cells and platelets and minimal harm to the recovered blood components and the recipient animal. The ATAS platform can process blood with a throughput range of $12 \mu\text{L}\cdot\text{min}^{-1}\sim 18 \mu\text{L}\cdot\text{min}^{-1}$, allowing the system to process the entire blood volume contained by an experimental mouse in ~ 2 hours. Utilizing the ATAS, we experimentally validated this claim and conducted the first therapeutic apheresis procedure on a sensitized mouse model, reducing preformed donor-specific antibodies and minimizing the risk of antibody-mediated rejection following organ transplantation.”

In the third paragraph of the *Discussion*, we added the following sentences:

“We will address this shortcoming in future prototypes with improvements in the design of the acoustic transducer, which will result in greater acoustic radiation forces and, hence, stronger separating power. Additionally, the fluctuation due to pulsatile flow can be further reduced by increasing the number of cavities in the fluid stabilizer. We are also exploring the possibility of incorporating three-dimensional standing acoustic waves generated in hybrid channels reported in our previous work ²³. This design has an acoustic intensity across the microfluidic channel that is several times

larger than the current device design, allowing for an increased throughput of 125 $\mu\text{L}\cdot\text{min}^{-1}$. This will significantly shorten the processing time by around tenfold.”

Reference

R1. Wu, M. et al. Circulating Tumor Cell Phenotyping via High-Throughput Acoustic Separation. *Small* 14, 1–10 (2018).

2. *Last paragraph of introduction: I suggest revising "limited blood volumes" in the first sentence to "small blood volumes".*

Our Response:

We appreciate the invaluable comments. Per the reviewer’s comment, we have modified the last paragraph of the Introduction accordingly, as well as in our results section where ‘limited blood volumes’ is utilized again.

Results:

1. *What is the anticoagulant used in the system?*

Our Response:

We appreciate the reviewer’s valuable question. The anticoagulant used in the system is Heparin sulfate (20 mL of blood/1 mL of heparin). Per the reviewer’s comment, we have added the information and detailed process in the Supplementary Materials.

2. *Are the antibodies removed simply as a result of the plasma being removed or does the device selectively pull out the antibodies from the whole blood? When plasma is removed, is a replacement fluid given to maintain blood volume?*

Our Response:

We appreciate the valuable questions. Antibodies are removed as a result of the plasma being removed since antibodies are soluble within the plasma. When the plasma is removed, an equal volume (~ 500ml) of replacement fluid, PBS, is provided with

cellular components for the mouse model. We have tested several common replacement fluids, such as PBS, 0.9% saline solution, 5% albumin solution, 5% dextrose solution and 10% dextrose solution, and presented the results in Supplementary Figure 7(b). The selection of replacement fluid has no significant impact on the separation efficiency.

To clarify the process, we have added the following descriptions in the revised manuscript. In the fifth and seventh paragraphs of “Constructing stable extracorporeal circulation and achieving highly efficient separation” section, we added the following sentences:

“Cellular components are separated from the blood and transferred into the sheath fluid, while antibodies remain in the plasma component due to their solubility.”

“...as depicted in Supplementary Fig. 8. It is worth noting that increasing the sheath flow from 60 $\mu\text{L}\cdot\text{min}^{-1}$ to 120 $\mu\text{L}\cdot\text{min}^{-1}$ maintains RBC and platelet recovery rates of ~95% and ~90% respectively. The sheath fluid was replaced with plasma to preserve blood volume after separation. We evaluated the effects of utilizing different sheath fluids, such as 5% albumin, 5% dextrose, PBS, 10% dextrose, and 0.9% saline solutions, and found the choice of fluid had a negligible effect on the recovery rates for RBCs and platelets.”

In the sixth paragraph of the “Characterization of ATAS using mouse models” section, we added the following sentences:

“We introduced anti-human CD3 monoclonal antibodies to mice whole blood and processed it with the ATAS system. In the ATAS system, cellular components were separated from the plasma and transferred to a replacement fluid, i.e., saline, to maintain blood volume. The plasma portion, as well as the immunoglobulins and other solvable antibodies, was not transferred. Thus, after processing by the ATAS system, the cellular components would not contain immunoglobulins.”

3. *What is the fluid balance of the subject after the procedure? Positive? Negative? Neutral?*

Our Response:

We appreciate the valuable questions. It was within euvoletic status. After the acoustic apheresis procedure, the plasma is removed while replacement fluid is given with cellular components of blood.

4. *What types of adverse effects can be expected/extrapolated when used in humans based on experience with mice or other small animals?*

Our Response:

We appreciate the valuable question. For human patients, the current device requires improvement to enhance its high throughput capabilities to process a larger volume of blood. In this case, a larger volume of anticoagulant may pose an issue when it returns with cellular components; however, heparin has been utilized and tolerated in humans for other devices, such as hemodialysis or continuous renal replacement therapy. The latter concern can be addressed by introducing a reversal agent such as protamine for heparin.

5. *If this device were to be used as a method for red cell exchange, could non-diseased red blood cells be infused as a replacement?*

Our Response:

We appreciate the valuable question. Our ATAS platform can separate cellular components and plasma, so it can be used as a method for red cell exchange. After removing the diseased red blood cells, the remaining plasma can be mixed with non-diseased red blood cells from other donors and then infused back into the patient's blood vessels. This process can be done based on our system by adding a mixing module after the separation module.

6. *What are the threshold values for lab parameters if applicable (eg hgb/hct, platelet count, etc) for it to be safe to put a subject on this device?*

Our Response:

We appreciate the valuable question. Technically, our ATAS platform can handle blood

samples with varied parameters. The difference in terms of platelet count or HGB level will not affect the performance of our platform. We will further investigate the safety threshold values for a subject and come up with a medical guideline. Per the reviewer's comments, we have added this point to the Discussion part as follows.

“Finally, the effects of repeated plasmapheresis on the host with our device are currently unknown and warrant further investigation. The safety threshold values (such as Hgb level, platelet count, etc.) must be investigated and verified for a subject to perform therapeutic apheresis on the ATAS platform.”

7. Is there a maximum cell count or antibody titer for this device to be effective? If yes, what are they? I am wondering if the system can get "saturated". Does efficiency vary according to the cell count? If so, what is the relationship between efficiency and lab parameters?

Our Response:

We appreciate the valuable questions. Our ATAS system is able to process whole blood samples without dilution. Thus, the cell count is $\sim 5 \times 10^9$ cells/mL. If the cell count is less, in other words, when blood is diluted, the separation performance can be improved. According to our previous work, diluted blood containing fewer blood cells can be separated successfully with high efficiency (>90%) and higher throughput (up to 200 $\mu\text{L}\cdot\text{min}$).^{R1}

Antibodies can be removed along with the plasma being removed if the antibodies are soluble in plasma. Therefore, we believe the antibody concentration will not have an impact on the efficiency.

Reference

R1. Wu, M. et al. High-throughput cell focusing and separation via acoustofluidic tweezers. *Lab on a Chip* 18, 3003-3010(2018).

Discussion

1. An explanation of how acoustics are able to separate cells from plasma would allow readers who have limited or no background in this field to appreciate the concept/idea more and make the manuscript more understandable.

Our Response:

We appreciate the invaluable comments. Per the reviewer's comments, we have added an extra figure to explain the mechanism of acoustics-based separation. The figure is inserted in Supplementary Fig. 1. The modification is as follows.

In the manuscript's second paragraph of the "Strategy and prototype of the ATAS" section, we added the following sentences:

"Acoustic pressure nodes are generated through the interference of two sets of surface acoustic waves produced by opposing pairs of interdigitated transducers. The acoustic radiation force deflects platelets and blood cells toward the acoustic pressure nodes, relocating them into the buffer. This process separates plasma and cellular components in real-time, resulting in an uninterrupted blood component separation. A detailed explanation of the acoustics-based separation mechanism is illustrated in Supplementary Fig. S1."

In the Supplementary Materials, we added the following sentences:

Supplementary Figure 1 | Schematic about the mechanism of acoustics-based separation.

2. *Discuss potential adverse effects on cells that acoustics have. Will it cause hemolysis which has a lot of downstream adverse consequences?*

Our Response:

We appreciate the invaluable comments. Per the reviewer's comments, we have investigated the morphology of blood components, P-selection level, hemoglobin level, and performed flow cytometry tests. From the results, we found no adverse effects on blood cells caused by acoustics.

Besides our work, some other researchers have also thoroughly examined the safety of acoustics-based blood cell separation for clinical applications. For example, Savage et al.^{R1} tested the blood parameters e.g., blood count, hemolysis, coagulation parameters, and platelet activation after processing by an acoustic apheresis microchannel. The results suggest adequate safety to blood cells to pursue additional studies. Specifically, they found less than 0.27% hemolysis due to the acoustics. This reference is also added to the citation.

Reference

R1. Savage W. et al. Safety of acoustic separation in plastic devices for extracorporeal blood processing. *Transfusion* 57, 1818-1826(2017).

Reviewer #2 (Remarks to the Author):

The manuscript describes therapeutic apheresis, and particularly focuses upon apheresis for small blood volumes in animal use, an interesting justification for acoustofluidics methods in this well-established technique. Overall, the work is sound and I can certainly see this contribution appearing in Nature Communications with its rather complete approach and compelling results, following all the way through to animal model use in relation to identifying and separating pathological antibodies in blood. Figure 4 makes it most clear, perhaps, that this method is a very interesting and powerful technique.

Our Response:

We appreciate the valuable feedback and time the reviewer devoted to evaluating our manuscript. We have prepared a point-by-point response to the comments and the revised manuscript. The changes have been highlighted in yellow for further review.

Comment:

I would encourage the authors to expand slightly on the idea of apheresis in mice, given the penchant the NIH has in funding and using murine models for nearly all types of scientific research. There are many scientific studies underway that rely on the mouse model as a consequence, arguably more than there should be given the limitations of mice in representing human health, but nonetheless mice are the dominant animal model in research. This contribution is especially useful in that context. Citation to a few acoustofluidics works in blood separation seems appropriate as well; for example, Zhang, et al. LoC 21:904-15 2021.

Our Response:

We appreciate your valuable feedback. As suggested by the reviewer, we have extended the scope of our ATAS platform to include all common types of laboratory animals. Per the reviewer's comments, we have also added some additional acoustofluidics works (including Zhang, et al. LoC 21:904-15 2021) to the citations.

Comment:

The frequency of the acoustic device appears to be missing, as does the rationale for the channel width and height versus attenuation length of the SAW in the substrate or perhaps the wavelength of the SAW and in the fluid. These aspects are important for the reader to understand the particle manipulation mechanism, at least as it is intended to be used by the authors in the work as reported.

Our Response:

We appreciate the valuable comments. Per the reviewer's comments, we have added the information in the Supplementary Materials and main text as follows.

“The spacing of interdigitated electrode is 50 μm , the length of the electrode is 10.5 mm, and the distance between two pairs of interdigitated electrodes is 9 mm.”

“The microfluidic channel for blood component separation has a width of 800 μm and a height of 100 μm .”

“The frequency of acoustic waves is ~ 20 MHz.”

Comment:

The introduction of the compressible air cavities is a nice solution to an annoying problem when using pulsatile flow pumps. The traditional solution is of course syringe pumps, yet they lack the ability to run for long periods and continually provide fluid. Noting the statement on p8, "With the assistance of the stabilizer, the recovery rates of RBCs and platelets can be increased to approximately 90% and 80%,..." is there anything further that could be done to improve these results? A second matching stage? Some may feel that greater separation fractions are needed in apheresis, and some thoughts on how that might be possible with some simple additions to your effort might be welcomed.

Our Response:

We appreciate the reviewer's valuable comments. The compressible air cavity can stabilize the pulsatile flow caused by peristaltic pumps. While the fluctuation can be reduced, it cannot be fully eliminated. This is one of the major reasons that prevent

further improvement in the recovery rates. Per the reviewer's suggestion, adding the number of cavities is a reasonable solution.

Per the reviewer's comments, we have added this point to the Discussion part as follows.

“Even with successful plasmapheresis in an *in vivo* sensitized mouse model, we found several limitations to the ATAS method. First, we observed incomplete separation of antibodies and cells (Fig. 3). Addressing this shortcoming would require improvements in the design of the acoustic transducer, which will result in greater acoustic radiation forces and, hence, stronger separating power. Besides, the fluctuation of pulsatile flow can be further reduced by adding the number of cavities in fluid stabilizer...”

Comment:

The ordering of the display items in Figure 1 (a-e) is a little odd; I don't know what I'd suggest to do differently, but it breaks the usual canon of left-to-right and top-to-bottom fairly significantly. If it can't be fixed so be it.

Our Response:

We appreciate the reviewer's valuable comments. Per the reviewer's comments, we have modified Figure 1 to follow the left-to-right and top-to-bottom canon. The new Figure 1 is as follows.

Fig. 1 | Strategy and prototype of the ATAS. **a.** The ATAS strategy aims to resolve the obstacles of apheresis in clinical and laboratory practices such as neonatal care, veterinary medicine, and scientific research. **b.** The ATAS strategy uses an acoustofluidic microchip where blood cells, platelets, and plasma are separated in a continuous and stable manner. **c.** Numerical simulation of fluid stabilizing performance. N represents the number of stabilizers. **d.** The acoustofluidic therapeutic apheresis prototype. **e.** Comparison map of the ATAS with other apparatus reported in literature. (i): ref. 17, (ii): ref. 44, (iii): ref. 45, (iv): ref. 46.

Comment:

p12: "through simple modifications to the chip design and working conditions, has the potential to separate various blood components,..." --- This should really be shown or rewritten a bit, because it is indeed difficult to isolate rare cells or RBCs from WBCs, and the text here gives the impression that any of these actions might be possible.

Our Response:

We appreciate the reviewer's valuable comments. Per the reviewer's comments, we

have rewritten the content. The revised paragraph is as follows.

“Besides, our device, through some modifications to the chip design and working conditions, has the potential to separate many types of blood components, such as red blood cells, platelets, lipoproteins, viruses, and extracellular vesicles. Although some challenges remain for isolating rare cells and separating red blood cells from white blood cells, the ATAS platform can pave the path to a wide range of biomedical applications.”

Comment:

p4: micron-level -> micro-scale

Our Response:

We appreciate the reviewer's valuable comments. The manuscript is revised accordingly.

Comment:

p5: singular blood steam

Our Response:

Thank you for pointing out this typo. We have revised it to “singular blood stream”.

REVIEWERS' COMMENTS

Reviewer #1 (Remarks to the Author):

This manuscript describes an acoustofluidic-based therapeutic apheresis system (ATAS) for performing plasmapheresis procedures in small-sized subjects. I have only 1 revision to recommend. There is a newer edition of the Guidelines for Therapeutic Apheresis (2023) so this should be the one referenced in reference #1 as opposed to the older one (2019 version).

Connelly-Smith L, Alquist CR, Aqui NA, Hofmann JC, Klingel R, Onwuemene OA, Patriquin CJ, Pham HP, Sanchez AP, Schneiderman J, Witt V, Zantek ND, Dunbar NM. Guidelines on the Use of Therapeutic Apheresis in Clinical Practice - Evidence-Based Approach from the Writing Committee of the American Society for Apheresis: The Ninth Special Issue. *J Clin Apher.* 2023 Apr;38(2):77-278. doi: 10.1002/jca.22043. PMID: 37017433.

Reviewer #2 (Remarks to the Author):

The authors have addressed my questions and concerns; I will be pleased to see the manuscript in print at Nature Communications.